# Seroprevalence and associated risk factors of *Toxoplasma gondii* infection among pregnant mother in Makassar, Indonesia

**Nurul Fadilah Ali Polanunu**[1,2☯], **Sitti Wahyuni**[1,3☯]*, **Firdaus Hamid**[4]

**1** Department of Parasitology, Faculty of Medicine, Hasanuddin University, Makassar, South Sulawesi, Indonesia, **2** Department of Parasitology, Faculty of Medicine, Universitas Muslim Indonesia, Makassar, South Sulawesi, Indonesia, **3** Department of Parasitology, Faculty of Medicine, University of Indonesia, Jakarta, Indonesia, **4** Department of Microbiology, Faculty of Medicine, Hasanuddin University, Makassar, South Sulawesi, Indonesia

☯ These authors contributed equally to this work.

* sittiwahyuni@gmail.com

**Data Availability Statement:** All relevant de-identified data are in the paper and its Supporting Information files.

## Abstract

The protozoan parasite, *Toxoplasma gondii* is estimated to infect one-third of the world's population. Infection in pregnant women can cause severe conditions for their babies. Until now, there is no data regarding *Toxoplasma* infection from Makassar pregnant mothers. This study aims to obtain information on *Toxoplasma* specific antibodies and to measure the risk factor associate with parasite infection. This cross-sectional study conducted in 9 of 47 primary health centres (Puskesmas) in Makassar. Blood samples and questionnaires were collected from 184 pregnant women aged 15–42 years old from September to October 2020. ELISA technique was used to examine the IgG and IgM antibodies. Univariable and multivariable analyses were carried out to measure factors that independently associate with *Toxoplasma* antibody positivity. Our result showed the range of *Toxoplasma* IgM and IgG are 0.06–1.01 and 0.09–3.01, respectively. While no one of our participants has an acute *Toxoplasma gondii* infection (IgM positive), we found 32,6% pregnant mothers are exposed to parasite (positive IgG). Contact with cats [OR(95%CI): 10.45(3.77–28.99)], consume chicken satay [OR(95%CI): 9.72(3.71–25.48)] and consume un-boiled water/ filtered water [OR(95%CI): 5.98(1.77–20.23)] are independently associate with positive *Toxoplasma* IgG antibody. Based on the result, we conclude that pregnant women in Makassar are exposed to *T. gondii* and the *oocyst* and *tissue cyst* of parasite contaminates food and water in Makassar.

## Introduction

*Toxoplasma gondii* is a protozoan parasite that infects humans and several warm blooded animals such as mammals and birds [1]. The definitive host of this parasite is feline and consuming food or drink contaminated by feline feces containing oocysts can cause toxoplasmosis [2]. In addition, infection also occurs through consuming undercooked meat which carries the

**Funding:** This study was supported by the Kementerian Riset Teknologi Dan Pendidikan Tinggi Republik Indonesia (grant no. 007/SP2H/LT/DRPM/2020 to SW).

**Competing interests:** No authors have competing interests

tissue *cyst* form of this protozoan [1]. These parasites can also be spread through sharing needles [1], blood transfusions [3], organ transplants [4] and from the mother to the fetus through the placenta [5] or when the baby passing the birth canal [4].

Acute *Toxoplasma* infection generally does not cause symptoms [6] but in immunocompromised infected individuals, toxoplasmosis can cause several clinical manifestations including neurologic symptoms [7]. The clinical manifestations of infants infected with *Toxoplasma* congenitally depend on what trimester the mother acquire the infection. Infection in the first trimester gives more severe symptoms than in the second and third trimesters [8]. Early infection can end with stillbirth or miscarriage, and if the baby survives, they may have serious problems like hydrocephalus, microcephaly, intracranial calcifications, retinochoroiditis, strabismus, blindness, epilepsy, psychomotor and mental retardation [4]. A child whose mother has *Toxoplasma* infection during pregnancy can be born normally and the symptoms appear years later [9].

A comprehensive systematic review and meta-analysis study done in 1,148,677 pregnant women from 91 countries reported the estimation of the global prevalence of latent toxoplasmosis was 33.8 and the higher prevalence was associated with low-income countries and low human development indices [10]. Three hundred thirty pregnant mothers from Indonesia also included in the analysis with seropositive prevalence was 10.9, which is lower than global prevalence [11]

Several risk factors are hypothesized to influence latent *Toxoplasma* infection. A study conducted in women of childbearing age found that living under unfavorable environmental conditions had approximately two times increased risk of being infected with the parasite [12]. Recently, a study from Central Java in Indonesia reported that IgG seropositive against *Toxoplasma* was associated with daily contact with raw meat, consuming unfiltered water and density of cats in the house [13].

Antenatal care is a routine examination during pregnancy that aimed to monitors fetal growth as well as to detect risk factors that may affect pregnancy and pregnancy outcome. The frequency of the antenatal care prompted by Indonesia Ministry of Health is 4 times, once in the first and the second trimesters and twice in the third trimester. In Indonesia, pregnant women from middle-high socioeconomic status commonly check their pregnancy in the private clinic or private hospital which is including early detection of TORCH (shorten from Toxoplasmosis, Other agents, Rubella, Cytomegalovirus and Herpes simplex) infections. Meanwhile, pregnant women from low socioeconomic status mostly visit primary health centers (Puskesmas) for antenatal care but without testing for TORCH. Service given in Puskesmas is free of charge. To date, there is no systematic data about infection or exposure of *Toxoplasma gondii* in Makassar, especially from pregnant women who attend government primary health centers.

## Material and methods

This study was conducted in Makassar, the capital city of South Sulawesi, Indonesia. The total area of Makassar is 199.3 $km^2$ and inhabited by 1.7 million people [14] with the number of pregnant women reported in 2019 being 30,936 (Makassar Development Planning Agency-unpublished data). In Indonesia, basic health services, including antenatal care are provided at primary health centers or Puskesmas. Each sub-district has at least one or more primary health centers that provide at least one doctor and one dentist to serve the community in the area. In Makassar, there are 47 Primary Health Centers spread across 14 sub-districts and 9 were selected randomly as our study sites. According to the Lemeshow formula, the number of pregnant women needs to be included in the study minimum of 188. Data were collected from

September until October 2020, one Puskesmas per week from Monday to Saturday, from 08.00 until 12.00 AM. The study proposal was reviewed by the Ethical Committee for Human Subject Research, Faculty of Medicine, Hasanuddin University and approved with ref number 8890 / UN4.24.1.2 / PT.01.04 / 2020. Verbal explanations regarding the purpose, benefits and research procedures were given to each pregnant mother and only those who gave signature on the approval form were included as study participants. Questions regarding participant demography, socioeconomic and several known Toxoplasmosis risk factors were asked directly to the participant. The information collection was performed by two research assistants that have been previously trained for the study purpose.

## Patient demography and *Toxoplasma* infection risk factors

Each participant was asked for age, history of gravidity and miscarriage as well as formal education and occupation. Risk factors for *Toxoplasma* infection were asked with questions related to non-medical risk factors. Contact with cat was asked through questions: Do you have cats? Do you allow cats to play inside your house? Do you have physical contact with cats (stroking, holding, contact with cat body parts? Do you clean cat litter? Do you have contact with soil? The question regarding water contamination is asked through the question: What is the source of the water used for washing dishes and tooth brushing)? For cooking preparation? For drinking? The question to find out food/drink contamination is asked through questions: During pregnancy, have you ever consumed undercooked meat (satay/steak)? Have you ever consumed raw vegetables? Have you ever consumed un-boiled water?

## Blood collection, storage and antibody measurement

A three ml blood from each participant was taken by using an EDTA tube and stored in the icebox in the standing position before sent to Parasitology laboratory, Hasanuddin University. To separate cells from plasma, after arriving in the laboratory the blood was centrifuged and the plasma was stored at minus 20˚C for further analysis. Antibody IgM and IgG levels for *Toxoplasma gondii* were measured by using kits from MyBioSource, Sandiego-California-USA, catalog numbers: MBS494234 for IgM and MBS494548 for IgG. The kit provides microwells coated *Toxoplasma*, enzyme conjugate, TBM substrate, stop solution, wash concentrate 20X solution, diluent sample, calibrator, buffer, positive control, negative control. The Enzyme-Linked Immunosorbent Assay (ELISA) was performed by following procedures from manufacture. For details, plasma samples from minus 20˚C were thawed at room temperature (25˚C) for 15 minutes and washing buffer (25 ml plus 475 ml distilled water) was prepared. Each 10 μL plasma was mixed with 200 μL sample diluent. A 100 μL of diluted plasma, calibrator, positive control and blank (contained sample dilution only) were pipetted into the well and incubated for 20 minutes at room temperature. After washing three times, 100 μL of the conjugated enzyme was added into each well and incubated for 20 minutes. A 100 μL TMB substrate was added into the well after washing and incubated for 10 minutes before adding a 100 μL stop solution. The binding was then read by the ELISA reader at wavelength 450 nm and the reading results were converted into antibody index by dividing each sample mean value by the cut-off value. The cut-off value was obtained by multiplying the calibrator mean and the calibrator factor (0.4 for IgM and IgG). Negative, borderline and positive *Toxoplasma* antibodies for both IgG and IgM were defined if the index value is <0.9, 0-9-1.1 and >1.1, respectively.

## Statistical analysis

Data taken from questionnaire results and the level of *Toxoplasma* antibody index value were analyzed using SPSS version 22 software package. The normality of continuous data was

checked by using a t-test and is presented by mean or median accordingly. Association between each risk factor with *Toxoplasma* antibody positivity was tested univariate by logistic regression analysis and all risk factors that shown P<0.1 then tested by multivariate analysis. Data presented as odds ratio and 95% confidence intervals.

## Result and discussion

### Participant characteristics and *Toxoplasma* specific antibodies

A total of two hundred pregnant mothers participated in this study. However only 184 were analyzed due to insufficient volume and the lysis of several blood samples. The age of the participants is 15–42 years old with an average of 26.9 years. There are 152 (82.6%) respondents who are not working to earn money (unemployed). Forty percent of participants have senior high school education background. Out of 184 respondents, there are 17.9% of mothers with a history of miscarriage. The plasma samples obtained were examined by using the ELISA technique to measure the level of antibody response against *T. gondii*. The range of *Toxoplasma* IgM antibody is 0.06–1.02 and no one samples were found to be positive for *T. gondii* IgM antibody. The positive *Toxoplasma* IgG antibody was detected in 32.61% of mothers. Characteristics of 184 pregnant mother and the details of antibody positivity to *Toxoplasma gondii* can be seen in Table 1.

The prevalence of *Toxoplasma* IgG in nine Puskesmas in Makassar vary between 27.02% to 50.00% (Table 2). This study found the highest prevalence of positive IgG *Toxoplasma* antibody is in pregnant mothers from Puskesmas Paccerakkang Biringkanaya sub-district and Puskesmas Mangasa, Rappocini sub-districtare 50.0%.

The number of pregnant mothers participates is vary between Puskesmas (5–37 participants), but the percentage (27–50%) of women with positive *Toxoplasma* IgG antibodies shown that none of the areas in Makassar are free from the parasites.

**Table 1. Characteristics of pregnant mother attending primary health centre (Puskesmas) for antenatal care in Makassar, Indonesia.**

| Characteristic | | n = 184 |
|---|---|---|
| Age, mean (SD) | | 26.97 (5.63) |
| Primigravida (%) | | 62(33.70) |
| Abortion history (%) | | 32 (17.90) |
| Education | | |
| | Elementary School (%) | 24 (13.04) |
| | Junior High School (%) | 31(16.85) |
| | Senior High School (%) | 87 (47.28) |
| | Diploma/ University (%) | 42 (22.83) |
| Employed (%) | | 32 (17.40) |
| *Toxoplasma* IgM Index, Range (Median) | | 0.06–1.02 (0.22) |
| *Toxoplasma* IgG Index, Range (Median) | | 0.09–3.00 (0.33) |
| *Toxoplasma* IgM Positivity | | |
| | Negative (%) | 178 (96.70) |
| | Positive (%) | 0 (0.00) |
| | Borderline (%) | 6 (3.30) |
| *Toxoplasma* IgG Positivity | | |
| | Negative (%) | 122 (66.30) |
| | Positive (%) | 60 (32.60) |
| | Borderline (%) | 2 (1.10) |

**Table 2. Distribution of positive *Toxoplasma* IgG antibody in Sub-districts in Makassar Indonesia.**

| Sub-district | Primary Health Center | Number of Sample | Positive IgG (%) |
|---|---|---|---|
| Biringkanaya | Sudiang | 22 | 36.36 |
| | Sudiang Raya | 5 | 40.00 |
| | Paccerakkang | 10 | 50.00 |
| Tallo | Kaluku Bodoa | 37 | 29.72 |
| Manggala | Antang | 37 | 27.02 |
| Rappocini | Kassi-Kassi | 25 | 32.00 |
| | Mangasa | 10 | 50.00 |
| Tamalate | Jongaya | 16 | 31.25 |
| | Tamalate | 22 | 27.27 |

### Specific *Toxoplasma* IgG antibody and risk factors

The interview data were juxtaposed with the ELISA results to identify which factors act as independent risk factors for *Toxoplasma* IgG antibody positivity. To avoid miscalculation, two borderline results were excluded from the analysis. A total of 13 factors were analyzed one by one by logistic regression to detect the factor that is attributable for positive specific *Toxoplasma* IgG antibody. From previous studies, age is one of the factors that influence immune response to *Toxoplasma* infections [15, 16]. In our study, we have checked the possibility of age as a confounding factor. Using Levene's test we found p = 0.387 indicated that the age of pregnant women in seropositive and seronegative are homogenous, therefore we did not include age as a confounding factor. There are 5 risk factors that associate to positive *T. gondii* IgG antibody, namely contact with cat [OR(95CI): 10.8(5.06–23.03)], drinking un-boiled water [OR(95%CI): 9.03(3.38–24.12)], consume undercook meat-chicken satay [OR(95%CI): 8.86(4.09–19.2)], consume un-boiled/ filtered water [OR(95%CI): 3.42(1.42–8.23], and contact with cat litter [COR(95%CI): 3.69 (1.35–10.07)]. Furthermore, to measure factors that independently influence *Toxoplasma* IgG antibody positivity, a multivariable logistic model was performed for variables that have p-value <0.1. In the analysis we found that contact with cats [OR (95%CI): 10.45(3.77–28.99)], consume undercook meat [OR(95%CI): 9.72(3.71–25.48)] and consume un-boiled/ filtered water [OR(95%CI): 5.98(1.77–20.23)] are independently associated with positive *Toxoplasma* IgG antibody. Univariate and multivariate analysis of *Toxoplasma* antibody risk factors can be seen in Table 3.

Active infection of a microbial can be seen from the increase of specific IgM antibody levels on serum or plasma. In the study we conducted at 9 primary health centers in Makassar we found that none of the 184 plasma samples had IgM antibodies against *Toxoplasma*. However, the antibody specific IgG can be detected from 32.6% of pregnant women which is not far from the percentage reported from Sanata Bamba, Burkina Faso (31.1%) [17] and Vueba, Angola 39.4% [5]. Similar study had been conducted 4 years ago visited 25 midwives in several primary health cares in Badung, Bali found the prevalence was 10.9% [11]. The presence of IgG antibodies against a microbial cannot be used as an indicator of infection, but from an epidemiology perspective, it might be important in figuring out the level of population exposures.

The risk factor of *Toxoplasma gondii* infection can be grouped into 3 categories: Immune status, medical contamination (sharing needle, transfusion, transplantation) and food/ drink (non-medical) contamination, in this study we are focusing on non-medic contaminations risk factors.

The risk of having positive *Toxoplasma*-IgG antibody increases 10 times if the mother has a history of contact with cat. Contact means direct contact like stroking and holding, and also

**Table 3. Risk factors associated with *Toxoplasma gondii* IgG antibody positivity among pregnant mother attending primary health center in Makassar, Indonesia.**

| Risk factors | | *Toxoplasma* IgG positivity | | Univariate | | Multivariate | |
|---|---|---|---|---|---|---|---|
| | | Positive N = 60 (%) | Negative N = 122 (%) | Crude Odds Ratio (95%CI) | *p*-value | Adjusted Odds Ratio (95%CI) | *p*-value |
| Gravidity | | | | | | | |
| | Multigravida | 42 (70) | 77 (63.10) | 1.36(0.7–2.65) | 0.360 | - | - |
| | Primigravida | 18 (30) | 45 (36.90) | 1 | | | |
| History of Miscarriage | | | | | | | |
| | Yes | 11 (18.30) | 19 (15.60) | 1.22(0.54–2.75) | 0.637 | - | - |
| | No | 49 (81.70) | 103 (84.40) | 1 | | | |
| House Member | | | | | | | |
| | >4 (Big) | 31 (51.70) | 57 (46.7) | 1.22(0.66–2.26) | 0.530 | - | - |
| | < = 4 (Small) | 29 (48.30) | 65 (53.30) | 1 | | | |
| Education Group | | | | | | | |
| | < = 9 Years | 15 (25.0) | 38 (31.10) | 0.74(0.37–1.48) | 0.39 | - | - |
| | >9 Years | 45 (75.0) | 84 (68.90) | 1 | | | |
| Employed | | | | | | | |
| | Yes | 14 (23.30) | 23 (18.90) | 1.31(0.62–2.78) | 0.481 | - | - |
| | No | 46 (76.70) | 99 (81.10) | 1 | | | |
| Water for Washing | | | | | | | |
| | Un-chlorinated | 33 (55.00) | 57 (46.70) | 1.39(0.75–2.60) | 0.294 | - | - |
| | Chlorinated | 27 (45.00) | 65 (53.30) | 1 | | | |
| Water for Cooking Preparation | | | | | | | |
| | Un-threated water | 22 (36.70) | 40 (32.80) | 1.19(0.62–2.27) | 0.604 | - | - |
| | Threated water | 38 (63.30) | 82 (67.20) | 1 | | | |
| Water for Drinking | | | | | | | |
| | Filtered | 53 (88.30) | 84 (68.90) | **3.42(1.42–8.23)** | **0.006** | 1.49(0.42–5.31) | 0.543 |
| | Boiled | 7 (11.70) | 38 (31.10) | 1 | | | |
| Raising Cat | | | | | | | |
| | Yes | 11 (18.30) | 13 (10.70) | 1.88 (0.79–4.45) | 0.155 | - | - |
| | No | 49 (81.70) | 109 (89.30) | 1 | | | |
| Contact with Cat | | | | | | | |
| | Yes | 35 (58.30) | 14 (11.50) | **10.8(5.06–23.03)** | **<0.001** | **10.45(3.77–28.99)** | **<0.001** |
| | No | 25 (41.70) | 108 (88.50) | 1 | | | |
| Cleaning Cat litter | | | | | | | |
| | Yes | 11 (18.30) | 7 (5.70) | **3.69(1.35–10.07)** | **0.011** | 0.78(0.20–3.03) | 0.714 |
| | No | 49 (81.70) | 115 (94.30) | 1 | | | |
| Contact with Soil | | | | | | | |
| | Yes | 12 (20) | 13 (10.70) | 2,10(0.89–4.93) | 0.090 | 0.84(0.28–2.53) | 0.751 |
| | No | 48 (80) | 109 (89.30) | 1 | | | |
| Consume undercook meat pregnancy | | | | | | | |
| | Yes | 50 (83.30) | 44 (36.10) | **8.86(4.09–19.20)** | **<0.001** | **9.72(3.71–25.48)** | **<0.001** |
| | No | 10 (16.70) | 78 (63.90) | 1 | | | |
| Consume raw vegetable during pregnancy | | | | | | | |
| | Yes | 37 (61.70) | 62 (50.80) | 1.45(0.78–2.72) | 0.244 | - | - |
| | No | 23 (38.30) | 60 (49.20) | 1 | | | |
| Consume un-boiled/ filtered water during pregnancy | | | | | | | |
| | Yes | 55 (91.70) | 67 (54.90) | **9.03(3.38–24.12)** | **<0.001** | **5.98(1.77–20.23)** | **0.004** |
| | No | 5 (8.30) | 55 (45.10) | 1 | | | |

contact with cat body parts such as fur, saliva and dry feces due to sharing a sofa or bed with a cat. A study done in primary school children in Southern Nigeria [18] and another study that carried out in Tunisia among pregnant women [19], found that there were no significant relationship between contact with cats and the seropositivity of *Toxoplasma gondii*. In those two studies there were no information about contact definition. Studies from South-Western Ethiopia [20] and North-Eastern China [21] reported that having cat at home increases risk of having positive *Toxoplasma* IgG antibody 5.82 times (95%CI: 1.61–20.99) and 2.55 times (95%CI: 1.31–4.97), respectively. In our population, we did not find that having cat at home is a risk factor to have parasite-specific IgG positive. This is probably because most Indonesians do not refuse visits from stray cats and there is a belief that driving away cats means rejecting good fortune.

The possibility of water as the media of *Toxoplasma oocyst* transmission was also examined in this study, namely source of water for washing dishes (include tooth brushing), cooking and drinking. From univariate analysis, we found that the filtered/ un-boiled water for drinking associate positively with positive *Toxoplasma* IgG, however, the significance disappears in multivariate analysis. Data from Taiz, Yemen reported that pregnant mothers who drink water from the surface area have higher *T. gondii* infections compared with the ones who drink tap water [22]. In Makassar, we found that the highest prevalence of *Toxoplasma* exposure was those who are living in areas without piped water access. When the pipe water is not available, people will prefer to drink filtered water. Filtered water is water purchased from a home industry sold in refill gallons available in many places in every subdistrict. The price per gallon is 5–7 times lower than the filtered water produced by branded companies. Because there is no supervision from the food and drug regulatory agency, the quality of the those cheap filtered water is unknown.

The *oocyst* of *Toxoplasma gondii* can enter the human body through consumption of contaminated raw vegetables or water while the tissue *cyst* is ingested undercook meat. To get an accurate information regarding consumption, we limit the consumption only during pregnancy. From the data we found consume undercook meat increases risk 9.7 times (OR, 95% CI = 3,71–25.48 to have positive *Toxoplasma* IgG. Meat in this study is chicken satay and not beef/goat/buffalo satay or steak. People from Makassar tend to consume chicken satay rather than beef which mostly serves as a soup or other dish that is processed with long time cooking. To ensure that chicken satay is undercook meat, three selling spots satay in different sub-districts in Makassar were surveyed and found that the length of burning time was vary between 3–6 minutes. The temperature of the satay once it was placed on the plate ranges from 40˚C to 51˚C and when the meat was cut open, the inside part was still juicy and reddish in color indicated the chicken satay was not well-done or undercooked. A study done a few decades ago reported that *Toxoplasma* cysts and fecal toxoplasma were killed after exposure to a temperature of 55˚C for 30 min [23] and according to the central for disease control (CDC), to avoid infection with tissue cysts from *Toxoplasma*, poultry meat should be cooked at a minimum temperature of 74˚C [1]. Although there is no report on *Toxoplasma* clinical manifestation in chicken, a molecular study conducted in Kenya reported that 79% of chicken farms infected with *Toxoplasma gondii* [24], suggesting that humans may get the infection through consuming undercook chicken. A study conducted in Senegal, West Africa reported that chicken farm consumes surface water have a higher prevalence of *Toxoplasma gondii* infection compared to chicken that drinks water from pipe or treated water [25]. A similar study from Nanjing, China [26] confirmed that free raise chicken is more vulnerable to be exposed to parasites compared to chicken from farms and it was related to the hygienist of the water they consume. In our data, we found that pregnant mothers who have history on drinking un-boiled/ filtered water have 5.9 times risk to have positive *Toxoplasma* IgG antibody. Similarly, a study carried

out in the rural areas of Yemen revealed that there was a significant association between drinking unimproved water with *T. gondii* infection among pregnant women [22]. Again from Northeastern Brazil, it was reported that seropositive *Toxoplasma gondii* associates with consuming ice from un-boiled water in [27]. All the data indicate the importance of contaminated water with the transmission of *Toxoplasma gondii*.

Although we found some factors are associate with Toxoplasma gondii IgG positivity, we realize that the range of 95% confidence intervals are wide indicating the sample size is small. During Covid-19 pandemic, the number of pregnant women who came to the Primary Health Centers decrease 50–70 than in normal situation. Initially, we collected 200 samples but 16 samples had to be excluded due to insufficient serum and blood lysis. Unfortunately, when we decided to add more participants, our city was locked-down due to an increase in the number of Covid-19 cases. It is unknown when the pandemic will end and we worried if delayed too long the quality of existing serum decreases, we then analyzed the existing samples whose numbers were not too far from the minimum sample size determined by the Lemshow formulas (n = 188). If the same study is to be repeated, it is recommended that the number of samples be increased.

## Conclusion

Overall, this study found that 32.6% of pregnant mother are exposed to *Toxoplasma gondii*. Consuming food or water that contaminated with *oocyst* or *tissue cysts* is important in parasite transmission in Makassar. To reduce the exposure of *Toxoplasma* in the population, the effort in increasing the food and drink hygienist need to be considered as a public health concern. To the extent of our knowledge, this study is the first study conducted in Makassar that measures exposure examine risk factors associate with parasites in term of food and water contamination.

## Supporting information

**S1 File.**
(SAV)

## Acknowledgments

The authors would like to thank the pregnant mothers who are willing to participate in the study and to Puskesmas staffs for helping us during sample collection.

## Author Contributions

**Conceptualization:** Sitti Wahyuni.

**Data curation:** Sitti Wahyuni.

**Investigation:** Nurul Fadilah Ali Polanunu.

**Methodology:** Nurul Fadilah Ali Polanunu, Firdaus Hamid.

**Supervision:** Sitti Wahyuni, Firdaus Hamid.

**Validation:** Sitti Wahyuni.

**Writing – original draft:** Nurul Fadilah Ali Polanunu.

**Writing – review & editing:** Sitti Wahyuni.

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
