## [Decision Letter · Decision Letter 0]

2 Feb 2021

PONE-D-20-41043

Seroprevalence and associated risk factors of Toxoplasma gondii infections among pregnant women in Makassar, Indonesia

PLOS ONE

Dear Dr. Wahyuni,

Thank you for submitting your manuscript to PLOS ONE. After careful consideration, we feel that it has merit but does not fully meet PLOS ONE’s publication criteria as it currently stands. Therefore, we invite you to submit a revised version of the manuscript that addresses the points raised during the review process.

Please refer to Reviewer #1 & #2: Major criticism: The manuscript entitled “Seroprevalence and associated risk factors of Toxoplasma gondii infections among pregnant women in Makassar, Indonesia” conducted a serological survey and risk factor analysis of Toxoplasmosis in Makassar, Indonesia. The results indicated that past infections T. gondii might associate with “contact with cats”, “un-boiled water”, and “satay consumption.” The finding could provide information of toxoplasmosis about non-medical risk factors in the study area.<br />Below are major issues1.     Due to the small sample size, the 95% confidence interval of the estimates in the models are quite wide, especially for those significant variables. Thus, the limitation should be discussed in the manuscript.2.     In addition to risk factors, what kind of confounding variables you have included the multivariate logistic regression?3.     Line 196-198. In fact, similar studies have been conducted in Swaziland and Nigeria. Please check those articles and modify the discussion accordingly.4.     Line 214-216. The risk chicken satay consumption could be elaborated more because satay is not undercooked meat. Could you show the data of different meat in the analysis? Why do you refer satay is “undercook meat?

We look forward to receiving your revised manuscript.

Kind regards,

Chia Kwung Fan, LL.M, PhD

Academic Editor

PLOS ONE

Journal Requirements:

"This study is funded by the Indonesian Ministry of Education."

"No"

3. We note that Figure 1 in your submission contain map images which may be copyrighted. All PLOS content is published under the Creative Commons Attribution License (CC BY 4.0), which means that the manuscript, images, and Supporting Information files will be freely available online, and any third party is permitted to access, download, copy, distribute, and use these materials in any way, even commercially, with proper attribution. For these reasons, we cannot publish previously copyrighted maps or satellite images created using proprietary data, such as Google software (Google Maps, Street View, and Earth). For more information, see our copyright guidelines: http://journals.plos.org/plosone/s/licenses-and-copyright.

3.1.    You may seek permission from the original copyright holder of Figure 1 to publish the content specifically under the CC BY 4.0 license. 

3.2.    If you are unable to obtain permission from the original copyright holder to publish these figures under the CC BY 4.0 license or if the copyright holder’s requirements are incompatible with the CC BY 4.0 license, please either i) remove the figure or ii) supply a replacement figure that complies with the CC BY 4.0 license. Please check copyright information on all replacement figures and update the figure caption with source information. If applicable, please specify in the figure caption text when a figure is similar but not identical to the original image and is therefore for illustrative purposes only.

4.Please upload a copy of Supporting Information Figure 1 which you refer to in your text (line 322).

Reviewers' comments:

Reviewer's Responses to Questions

**Comments to the Author**

1. Is the manuscript technically sound, and do the data support the conclusions?

Reviewer #1: Yes

Reviewer #2: Yes

2. Has the statistical analysis been performed appropriately and rigorously? 

Reviewer #1: Yes

Reviewer #2: Yes

3. Have the authors made all data underlying the findings in their manuscript fully available?

Reviewer #1: Yes

Reviewer #2: Yes

4. Is the manuscript presented in an intelligible fashion and written in standard English?

Reviewer #1: No

Reviewer #2: Yes

5. Review Comments to the Author

Reviewer #1: This study intends to detect the seroprevalence and associated risk factors of Toxoplasma gondii infections among

pregnant women in Makassar, Indonesia and their findings are important to the policy-makers to delineate a preventive measures against Toxoplasma infection for women. The results are appropriately justified by the methods. However, a specific term should be clarified throughout the contents, saying oocyst or tissue cyst. In their study , they claimed "the cyst of parasite contaminates food and water in Makassar.". Do you mean oocyst or tissue cyst?

Reviewer #2: The manuscript entitled “Seroprevalence and associated risk factors of Toxoplasma gondii infections among pregnant women in Makassar, Indonesia” conducted a serological survey and risk factor analysis of Toxoplasmosis in Makassar, Indonesia. The results indicated that past infections T. gondii might associate with “contact with cats”, “un-boiled water”, and “satay consumption.” The finding could provide information of toxoplasmosis about non-medical risk factors in the study area.

Below are major issues

1. Due to the small sample size, the 95% confidence interval of the estimates in the models are quite wide, especially for those significant variables. Thus, the limitation should be discussed in the manuscript.

2. In addition to risk factors, what kind of confounding variables you have included the multivariate logistic regression?

3. Line 196-198. In fact, similar studies have been conducted in Swaziland and Nigeria. Please check those articles and modify the discussion accordingly.

4. Line 214-216. The risk chicken satay consumption could be elaborated more because satay is not undercooked meat. Could you show the data of different meat in the analysis? Why do you refer satay is “undercook meat?

6. PLOS authors have the option to publish the peer review history of their article (what does this mean?). If published, this will include your full peer review and any attached files.

Reviewer #1: No

Reviewer #2: No

---

## [Author Response · Author response to Decision Letter 0]

19 Apr 2021

Makassar, March 16th 2021

Dear Academic Editor and Reviewer (s),

We have made improvements to the manuscript by following the recommendations of the reviewers. These improvements are as follows: 

1. Although we found some factors are associate with Toxoplasma gondii IgG positivity, we realize that the range of 95% confidence intervals are wide indicating the sample size is small. During Covid-19 pandemic, the number of pregnant women who came to the Primary Health Centers decrease 50-70 than in normal situation. Initially, we collected 200 samples but 16 samples had to be excluded due to insufficient serum and blood lysis. Unfortunately, when we decided to add more participants, our city was locked-down due to an increase in the number of Covid-19 cases. It is unknown when the pandemic will end and we worried if delayed too long the quality of existing serum decreases, we then analyzed the existing samples whose numbers were not too far from the minimum sample size determined by the Lemshow formulas (n=188). If the same studi is to be repeated, it is recommended that the number of samples be increased (line 252-261).

2. From previous studies, age is one of the factors that influence immune response to Toxoplasma infections. In our study, we have checked the possibility of age as a confounding factor. Using Levene's test we found p=0.387 indicated that the age of pregnant women in seropositive and seronegative are homogenous, therefore we did not include age as a confounding factor (line 171-174).

3. We have read the recommended paper and we found several information:

a. Paper from Nigeria written by Gyang, et al. conducted a study of seroprevalence and associated risk factors among primary schoolchildren. Among the risk factors tested including contact with cats, none showed statistical significance associated. The paper has a different object than us and did not explain about what kind of contact with cat it is, whether it is just stroking or sleeping together. However, in our paper, we have mentioned that contact with cat means not only physical contacts like stroking or carrying but also contact with body part of cat.

b. Another Research conducted in Swaziland by Liao, et al. explained that seropositivity among schoolchildren was around 8% positive for Toxoplasma gondii. However, this paper did not discuss the association of contact with cats with exposure to Toxoplasma gondii. They discussed about the risk factors like gender and age group among the children. However, we found another study carried out in Tunisia that investigated the association between contact with cat and seropositivity of Toxoplasma gondii in pregnant women. They found that contact with cat was independently associated with positivity of Toxoplasma gondii antibody. 

The paper mentioned above were included in discussion (line 202-205)

4. During the interview, we have confirmed that the satay in this study refers to chicken satay and not beef / goat / buffalo satay or steak. People of Makassar tend to consume chicken satay rather than beef which mostly serves as a soup or other dish that is processed with long time cooking. Furthermore, Indonesian prefer to consume chicken satay that is not too dry (undercooked) to avoid tough meat and lack of taste. To ensure that chicken satay is undercook meat, three selling spots satay in different sub-districts in Makassar were surveyed and found that the length of burning time was vary between 3-6 minutes. The temperature of the satay once it is placed on the plate ranges from 40oC to 51oC and when the meat was cut open, the inside part was still juicy and reddish in colour indicated the chicken satay was not well-done or undercooked. A study done a few decades ago reported that Toxoplasma cysts and faecal toxoplasma were killed after exposure to a temperature of 55oC for 30 min and according to the central for disease control (CDC), to avoid infection with tissue cysts from Toxoplasma, poultry meat should be cooked at a minimum temperature of 74oC (line 229-237)

We hope that revision we propose meet the PLOS ONE’s publication criteria

Yours sincerely,

Sitti Wahyuni

---

## [Decision Letter · Decision Letter 1]

3 May 2021

Seroprevalence and associated risk factors of Toxoplasma gondii infection among pregnant mother in Makassar, Indonesia

PONE-D-20-41043R1

Dear Dr. Wahyuni,

We’re pleased to inform you that your manuscript has been judged scientifically suitable for publication and will be formally accepted for publication once it meets all outstanding technical requirements.

Kind regards,

Chia Kwung Fan, LL.M, PhD

Academic Editor

PLOS ONE

Additional Editor Comments (optional):

Reviewers' comments:

Reviewer's Responses to Questions

**Comments to the Author**

1. If the authors have adequately addressed your comments raised in a previous round of review and you feel that this manuscript is now acceptable for publication, you may indicate that here to bypass the “Comments to the Author” section, enter your conflict of interest statement in the “Confidential to Editor” section, and submit your "Accept" recommendation.

Reviewer #1: All comments have been addressed

Reviewer #2: All comments have been addressed

2. Is the manuscript technically sound, and do the data support the conclusions?

Reviewer #1: Yes

Reviewer #2: Yes

3. Has the statistical analysis been performed appropriately and rigorously? 

Reviewer #1: Yes

Reviewer #2: Yes

4. Have the authors made all data underlying the findings in their manuscript fully available?

Reviewer #1: Yes

Reviewer #2: Yes

5. Is the manuscript presented in an intelligible fashion and written in standard English?

Reviewer #1: Yes

Reviewer #2: Yes

6. Review Comments to the Author

Reviewer #1: (No Response)

Reviewer #2: All the comments have been addressed by the authors. The manuscript is ready for publication for the current version.

7. PLOS authors have the option to publish the peer review history of their article (what does this mean?). If published, this will include your full peer review and any attached files.

Reviewer #1: No

Reviewer #2: No

---

## [Editor Report · Acceptance letter]

26 May 2021

PONE-D-20-41043R1 

Seroprevalence and associated risk factors of *Toxoplasma gondii* infection among pregnant mother in Makassar, Indonesia 

Dear Dr. Wahyuni:

I'm pleased to inform you that your manuscript has been deemed suitable for publication in PLOS ONE. Congratulations! Your manuscript is now with our production department. 

Kind regards, 

on behalf of

Dr. Chia Kwung Fan 

Academic Editor

PLOS ONE